# In Search for a SAT-friendly Binarized Neural Network Architecture

**Nina Narodytska**
VMware Research
Palo Alto, USA
`nnarodytska@vmware.com`

**Hongce Zhang**[*]
Princeton University
Princeton, USA
`hongcez@princeton.edu`

**Aarti Gupta**
Princeton University
Princeton, USA
`aartig@princeton.edu`

**Toby Walsh**
UNSW Sydney, Data61
Sydney, Australia
`tw@cse.unsw.edu.au`

## Abstract

Analyzing the behavior of neural networks is one of the most pressing challenges in deep learning. Binarized Neural Networks are an important class of networks that allow equivalent representation in Boolean logic and can be analyzed formally with logic-based reasoning tools like SAT solvers. Such tools can be used to answer existential and probabilistic queries about the network, perform explanation generation, etc. However, the main bottleneck for all methods is their ability to reason about large BNNs efficiently. In this work, we analyze architectural design choices of BNNs and discuss how they affect the performance of logic-based reasoners. We propose changes to the BNN architecture and the training procedure to get a simpler network for SAT solvers without sacrificing accuracy on the primary task. Our experimental results demonstrate that our approach scales to larger deep neural networks compared to existing work for existential and probabilistic queries, leading to significant speed ups on all tested datasets.

## 1 Introduction

Deep neural networks are among the most successful AI technologies making impact in a variety of practical applications ranging from vision to speech recognition and natural language (Goodfellow et al., 2016). However, many concerns have been raised about the decision making process behind machine learning technology. For instance, can we trust decisions that neural networks make (EU Data Protection Regulation, 2016; Goodman & Flaxman, 2017; NIPS IML Symposium, 2017)? One way to address this problem is to define properties that we expect the network to satisfy. Verifying whether the network satisfies these properties sheds light on the properties of the function that it represents. Verification guarantees can reassure the user that the network behaves as expected.

There are two main approaches to neural network analysis. The first approach, the *certification* of neural networks, trains a verified network that satisfies given properties, e.g. a network that is guaranteed to be robust to adversarial perturbations (Wong & Kolter, 2018; Dvijotham et al., 2018; Raghunathan et al., 2018; Mirman et al., 2018). However, a set of properties must be known in advance, which might not always be possible. Moreover, enforcing a set of properties during the training procedure can significantly affect the accuracy of the network on the primary task. Finally, certification techniques work with relaxation of the original problem and might not be able to certify robust inputs. The second approach, the *verification* of neural networks, takes a trained network as input and focuses only on the verification task (Katz et al., 2017; Weng et al., 2018; Singh et al., 2019; Xiao et al., 2019). A number of verification frameworks were proposed over the last few years that can be roughly divided into complete (Katz et al., 2017; 2019; Tjeng et al., 2019) and incomplete methods (Weng et al., 2018; Zhang et al., 2018; Singh et al., 2018). As the training and

---

[*]This work was mostly done during internship at VMware Research.

verification tasks are separated, a wide set of properties can be checked. However, the scalability of this approach remains an issue, especially for complete methods.

In this work we tackle the scalability problem of the complete verification approach, in the context of an important class of networks, Binarized Neural Networks(BNNs) (Hubara et al., 2016). A BNN is an extreme case of quantized neural networks where parameters are primarily binary. These networks have a number of important features that are useful in resource constrained environments, like embedded devices or mobile phones (McDanel et al., 2017; Kung et al., 2017). They are memory efficient as only one bit per weight must be stored and are computationally efficient as all activations are binary, which enables the use of specialized algorithms for fast binary matrix multiplication. More importantly, these networks admit an exact representation in Boolean logic (Narodytska et al., 2018; Cheng et al., 2018). Such a representation enables us to apply powerful Boolean reasoning tools to the analysis of BNNs. For example, we can perform a rich set of queries, ranging from existential queries (e.g., is there a faulty input to the network?), to counting queries (e.g., how many faulty inputs exist?), using logic-based reasoners (Baluta et al., 2019; Shih et al., 2019).

This paper makes two main contributions.

- First, we analyze how different architectural design choices for BNNs affect the performance of SAT solvers. To identify influential parts of the design, we scrutinize BNNs at three levels of granularity: individual neurons, blocks of layers, and network as a whole. Our work continues work of (Narodytska et al., 2018; Cheng et al., 2018) where BNNs were analyzed on the network level. For the block and network levels, we only analyse bottlenecks, propose possible research directions, and position existing work on the network level. Our main contribution here is within the network level.

- Second, we exploit our findings to train SAT-friendly BNNs. We propose a modified training procedure that makes the resulting network easier for logic-based verification tools, like SAT solvers, to reason about. Modifications to training are crucially performed so that the accuracy of the network is unaffected. Overall, our approach (a) preserves the separation between training and verification, by not committing to a certain property during training and (b) boosts the performance of logic-based verification. We implemented the proposed methods and demonstrated significant performance gains over previous work (Narodytska et al., 2018; Khalil et al., 2019; Baluta et al., 2019). We get more than 10x-20x improvements on tested benchmarks for both verification and quantitative queries, e.g., finding the probability that a perturbation yields an adversarial example.

## 2 BACKGROUND

**Boolean satisfiability (SAT).** We assume notation and definitions standard in Boolean Satisfiability (SAT), i.e. the decision problem for propositional logic (Biere et al., 2009). SAT formulas are defined over a set of Boolean variables $\{\mathbf{x}_1, \ldots, \mathbf{x}_n\}$. A literal $l_i$ is a variable $\mathbf{x}_i$ ('positive' polarity) or its complement $\bar{\mathbf{x}}_i$ ('negative' polarity). *Cardinality constraints* over Boolean variable are constraints of the form $\sum_{i=0}^{n} l_i \geq k$, where $l_i \in \{\mathbf{x}_i, \bar{\mathbf{x}}_i\}$. In this work we employ *reified cardinality constraints*: $\mathbf{x}' \Leftrightarrow \sum_{i=0}^{n} l_i \geq k$. We use the full sequential counters encoding (Sinz, 2005b) to model this constraint in SAT (see Appendix A for details of the encoding). However, other encodings can be used. We also work with (approximate) solutions counting solvers, e.g. ApproxMC3. See details on such methods in Appendix as well A.

**Binarized neural networks** We summarise standard BNN structure (Hubara et al., 2016). A BNN consists of blocks of layers, each mapping binary vectors to binary vectors. We define a *block* (referred to as BLOCK) as a function mapping an input to an output, i.e., $\text{BLOCK}^i : \{-1, 1\}^{n_i} \rightarrow \{-1, 1\}^{n_{i+1}}$, $i = 1, \ldots, d - 1$, where $d - 1$ is the number of hidden layers. The last block, denoted OUTPUT, has a slightly different structure: $\text{OUTPUT} : \{-1, 1\}^{n_d} \rightarrow \mathbb{R}^p$, where $p$ is the number of outputs of the network. Each BLOCK takes an input vector $x$ and applies three transformations: a linear transformation (LIN), batch normalization (BN) and binarization (BIN)[1]. Table 1 shows transformations of internal and output blocks in detail.

---

[1]In the training phase, there is an additional *hard* $\tanh$ layer after batch normalization, but it is redundant in the inference phase.

| Structure of $i$th internal block, $\text{BLOCK}_i : \{-1,1\}^{n_i} \to \{-1,1\}^{n_{i+1}}$ on input $x^i \in \{-1,1\}^{n_i}$ | |
|---|---|
| LIN | $y = A^i x^i + b^i$ , where $A^i \in \{-1,1\}^{n_{i+1} \times n_i}$ and $y, b^i \in \mathbb{R}^{n_{i+1}}$ |
| BN | $z_j = \alpha_j^i \left( \frac{y_j - \mu_j^i}{\sigma_j^i} \right) + \gamma_j^i$, where $\alpha^i, \gamma^i, \mu^i, \sigma^i, z \in \mathbb{R}^{n_{i+1}}$ |
| BIN | $x^{i+1} = \text{sign}(z)$, where $x^{i+1} \in \{-1,1\}^{n_{i+1}}$ |

| Structure of the output block, $\text{OUTPUT} : \{-1,1\}^{n_d} \to \mathbb{R}^p$ on input $x^d \in \{-1,1\}^{n_d}$ | |
|---|---|
| LIN | $o = C x^d + q$, where $C \in \{-1,1\}^{p \times n_d}$ and $q \in \mathbb{R}^p$ |

Table 1: Structure of internal and output blocks, which, stacked together, form a BNN.

## 3 LOGIC-BASED ANALYSIS OF BNNs

An important property of BNNs is that they allow exact encoding into SAT: for each BNN there exists a SAT formula such that solutions of the formula are exactly the set of all valid input/output pairs of the BNN (Narodytska et al., 2018; Cheng et al., 2018). Such translation opens many possibilities for logic-based analysis of BNNs. In this section, we describe several such methods. We highlight that reasoning tools available for BNNs are more diverse and powerful compared to frameworks available for the analysis of conventional networks, which focus mostly on verification and certification of NNs. Since logic-based analysis of BNNs exploits their logical representation, we start by revisiting how to obtain such a representation before discussing logic-based BNN analysis tools.

### 3.1 SAT ENCODING OF BNNs

We consider how to generate a logical encoding of a BNN (Narodytska et al., 2018). Consider a single block $\text{BLOCK}^i$, which applies three transformations to the input vector. The first step is a linear (affine) transformation (LIN) of the input vector. The linear transformation can be based on a fully connected layer or a convolutional layer. The linear transformation is followed by a scaling which is performed with a batch normalization operation (BN) (Ioffe & Szegedy, 2015a). Finally, a binarization is performed using the sign function to obtain a binary output vector (BIN). We will use the following running example to demonstrate the encoding.

**Example 3.1.** *Consider a block with three inputs and two outputs. We define transformation parameters as follows:*

$$\text{LIN}:$$
$$A = [1, -1, 1; -1, -1, 1], b = [0, 0]$$
$$\text{BN}:$$
$$\alpha = [0.1, 0.1], \mu = [1, -1], \sigma = [1, 1], \gamma = [0.1, 0.2] \quad \square$$

We encode these transformations as a system of logical constraints. We introduce a vector of binary variables $x^i \in \{-1,1\}^{n_i}$ to encode inputs and $x^{i+1} \in \{-1,1\}^{n_{i+1}}$ to encode outputs of $\text{BLOCK}^i$. Since transformations are applied sequentially, we can encode the block as their composition:

$$z_j = \frac{\alpha_j^i}{\sigma_j^i} (\sum_{t=1}^{n_i} a_{j,t}^i x_t^i + b_j^i - \mu_j^i) + \gamma_j^i, \qquad\qquad j = 1 \ldots n_{i+1} \qquad\qquad (1)$$

$$x_j^{i+1} = \text{sign}(z_j), \qquad\qquad j = 1 \ldots n_{i+1}, \qquad\qquad (2)$$

where $a_{j,t}^i$ and $b_j^i$ are parameters of LIN, $\alpha_j^i, \sigma_j^i, \mu_j^i$ and $\gamma_j^i$ are parameters of BN. To obtain a SAT encoding, we need to eliminate binary variables as a SAT formula is defined over Boolean variables. We introduce Boolean variables $\mathbf{x}_t \in \{0,1\}$, $t = 1, \ldots, n_i$ and relate them to the corresponding binary variables $x_t$: $x_t = 2\mathbf{x}_t - 1$, $t = 1, \ldots, n_i$. We get that if $\mathbf{x}_t = 0$ then $x_t = -1$, otherwise $x_t = 1$. We can now rewrite the last expression as

$$\mathbf{x}_j^{i+1} = \text{sign}\left( \frac{\alpha_j^i}{\sigma_j^i} (\sum_{t=1}^{n_i} a_{j,t}^i 2\mathbf{x}_t^i - \sum_{t=1}^{n_i} a_{j,t}^i + b_j^i - \mu_j^i) + \gamma_j^i \right), j = 1 \ldots n_{i+1}.$$

We denote $C_j = (-\sum_{t=1}^{n_i} a_{j,t}^i + b_j^i - \mu_j^i)$, as these are just constants. We get

$$\mathbf{x}_j^{i+1} \Leftrightarrow \begin{cases} \sum_{t=1}^{n_i} a_{j,t}^i \mathbf{x}_t^i \geq \frac{-\gamma_j^i \sigma_j^i}{2\alpha_j^i} - \frac{C_j}{2}, & \text{if } \alpha_j^i > 0 \\ \sum_{t=1}^{n_i} a_{j,t}^i \mathbf{x}_t^i \leq \frac{-\gamma_j^i \sigma_j^i}{2\alpha_j^i} - \frac{C_j}{2}, & \text{if } \alpha_j^i < 0 \\ \gamma_j \geq 0, & \text{if } \alpha_j^i = 0. \end{cases} \quad , j = 1 \ldots n_{i+1}, \quad (3)$$

**Example 3.2.** *Consider how the encoding works on Example 3.1. We introduce three binary variables to encode inputs $x_i^0$, $i = 0, 1, 2$, and two binary variables to encode outputs $x_0^1$ and $x_1^1$. Inputs and outputs are connected as follows:*

$$x_0^1 = \text{sign}(0.1(x_0^0 - x_1^0 + x_2^0 - 1) + 0.1 \geq 0);$$
$$x_1^1 = \text{sign}(0.1(-x_0^0 - x_1^0 + x_2^0 + 1) + 0.2 \geq 0).$$

*Next we perform variable replacement as explained above using the relation $x_t = 2\mathbf{x}_t - 1$. Note that $\alpha_1^0$ and $\alpha_2^0$ are both positive so we get:*

$$\mathbf{x}_0^1 \Leftrightarrow (0.1(2\mathbf{x}_0^0 - 1 - (2\mathbf{x}_1^0 - 1) + 2\mathbf{x}_2^0 - 1 - 1) + 0.1 \geq 0) \Leftrightarrow (\mathbf{x}_0^0 - \mathbf{x}_1^0 + \mathbf{x}_2^0 \geq 1/2),$$

$$\mathbf{x}_1^1 \Leftrightarrow (0.1(-(2\mathbf{x}_0^0 - 1) - (2\mathbf{x}_1^0 - 1) + 2\mathbf{x}_2^0 - 1 + 1) + 0.2 \geq 0) \Leftrightarrow (-\mathbf{x}_0^0 - \mathbf{x}_1^0 + \mathbf{x}_2^0 \geq -4/2). \quad \square$$

The final step is to encode reified cardinality constraints using Boolean variables instead of binary (integer) variables in Equation 3. We recall the following tautology for a Boolean variable $\mathbf{x}_t + \bar{\mathbf{x}}_t = 1$, where $\bar{\mathbf{x}}_t$ is a negation of $\mathbf{x}_t$. As all coefficients $a_{jt}^i \in \{-1, 1\}$, Equation 3 contains linear constraints with unary coefficients (see the last transformation in Example 3.2). We denote $N_j = |\{a_{jt}^i < 0, t = 1, \ldots, n_{i+1}\}|$ the number of negative coefficients in the $j$th row and assume $\alpha_j^i \neq 0$.

$$\mathbf{x}_j^{i+1} \Leftrightarrow (l_1^i + \ldots + l_{n_i}^i \geq h_j), \quad (4)$$

$$\text{where } l_t^i = \begin{cases} \mathbf{x}_t^i & \text{if } a_{jt}^i > 0 \\ \bar{\mathbf{x}}_t^i & \text{if } a_{jt}^i < 0 \end{cases} \text{ and } h_j = \begin{cases} \lceil -\gamma_j^i \sigma_j^i/(2\alpha_j^i) - C_j/2 - N_j \rceil & \text{if } \alpha_j^i > 0 \\ \lfloor -\gamma_j^i \sigma_j^i/(2\alpha_j^i) - C_j/2 - N_j \rfloor & \text{if } \alpha_j^i < 0. \end{cases}$$

We recall that a constraint of type Equation 4 is a *reified cardinality constraint* that can be translated into Boolean formulae (Narodytska et al., 2018). We present details of such translation in Appendix A. We refer to a SAT encoding of BLOCK as $\text{BINBLOCK}^i(\mathbf{x}^i, \mathbf{x}^{i+1})$. Similarly, we can encode the OUTPUT block that we show in details below as our method needs to make modifications to the original encoding (see Appendix B.1). We refer to encoding of OUTPUT as BINOUTPUT. We can now represent the entire network as a Boolean formula: $\text{BINBNN}(\mathbf{x}, o) \equiv \bigwedge_{i=1}^{d-1} \text{BINBLOCK}^i(\mathbf{x}^i, \mathbf{x}^{i+1}) \wedge \text{BINOUTPUT}(\mathbf{x}^d, o)$, where $\text{BINBLOCK}^i(\mathbf{x}^i, \mathbf{x}^{i+1})$ encodes the $i$th block $\text{BLOCK}^i$ with input $\mathbf{x}^i$ and output $\mathbf{x}^{i+1}$, $i \in [0, d-1]$, $\text{BINOUTPUT}(\mathbf{x}^d, o)$ is a Boolean encoding of the last layer.

## 3.2 BNN'S REASONERS

We consider three types of reasoners designed for BNNs. The first type includes property checkers that can handle existential queries, e.g. 'is there an input of the network that violates a given property?'. The second type deals with probabilistic queries, e.g. 'what is an approximate probability that a valid input will violate a given property?'. Finally, the most powerful reasoner answers a large set of queries, like generating explanations, finding exact probabilities of property violations, etc.

**Property checkers.** Property checking of BNNs using a SAT-based approach was proposed in (Narodytska et al., 2018; Cheng et al., 2018). Given a precondition $prec$ on the inputs $\mathbf{x}$ and a property $prop$ on the outputs $o$, we check if the following statement is valid:

$$prec(\mathbf{x}) \wedge \text{BINBNN}(\mathbf{x}, o) \Rightarrow prop(o).$$

To decide if there exists a counterexample to this property, we look for a model of:

$$prec(\mathbf{x}) \wedge \text{BINBNN}(\mathbf{x}, o) \wedge \neg prop(o).$$

An example of property checking is to check for the existence of adversarial perturbations. In this case, $prec$ defines an $\epsilon$-ball of valid perturbations and $prop$ states that the classification should not change under small perturbations. An optimization version of this problem was considered in (Khalil et al., 2019) using ILP rather than SAT solvers to reason about BNNs.

**Quantitative reasoners.** In many practical applications, it is not sufficient to check for the existence of a counterexample to a given property. We would like to know precise or approximate probability of the undesired behavior. (Baluta et al., 2019), proposes a framework to answer such probabilistic queries using approximate model counting tools. The main technical challenge is to perform efficient approximate solution counting of a SAT formula that contains a BINBNN as a subformula. Consider the property verification formula above. By approximately counting solutions, we obtain an estimate of the probability of a valid input leading to a property violation with a controllable and bounded error [2]. Baluta et al. (2019) identified three applications of this framework in the security domain: robustness, trojan attacks, and fairness. Narodytska et al. (2019) investigated how a similar type of quantitative reasoning can be used to assess the quality of ML explanations.

**Knowledge compilation engines.** The compilation of BNNs is an interesting research direction that aims to compile a BINBNN into a tractable structure, a logic graph-based representation of the formula (Shih et al., 2019; Choi et al., 2019) that supports a wide range of queries about the original BNN, including exact solution counting for probabilistic queries or generating logical explanations for network decisions. Interestingly, BNNs often admit succinct representations as the networks contain redundancies that can be eliminated by compilation. For example, (Choi et al., 2019) reports large reduction in the representation size on some benchmarks.

As can be seen from this overview, the success of these applications depends on the ability to reason efficiently about the underlying formula in the corresponding logical reasoner, e.g., a SAT solver, an approximate model counting method, or a knowledge compilation engine. On the one hand, BNNs are potentially easier to reason about compared to full-precision networks [3]. On the other hand, SAT is a hard combinatorial problem. Hence, developing efficient decision procedures require exploiting the special structure of BNNs.

## 4 ANALYSIS OF BNN'S FROM THE SAT STANDPOINT

In this section we analyze the properties of the formula BINBNN from the perspective of a solver developer. We discuss how these properties affect solver performance both positively and negatively. We highlight parts of the BNN architecture that can be exploited to speed up logical reasoners. To structure our analysis, we consider BNN's at three levels of granularity that naturally correspond to sub-formulas of BINBNN. (Narodytska et al., 2018; Cheng et al., 2018; Khalil et al., 2019) discussed and exploited properties of BNNs on the highest level of granularity, but we believe that there are more opportunities to take advantage of the structure if we look at all levels systematically.

**Neuron level** We start with the level of an individual neuron. Recall that the value of a neuron is determined by Equation 4. Let us examine it in more detail. We note that $A$ is a *dense matrix* by design as all entries must be $1$ or $-1$. At the same time, the number of literals in Equation 4 is equal to the width of $A$ for fully-connected layers or the filter size for convolutional layers. Therefore, the high density of $A$ leads to a large number of variables in the cardinality constraint of Equation 4. To make things worse, depending on the encoding, the number of auxiliary variables to encode Equation 4 depends on $n_i$ and $h_j$, e.g., the number of auxiliary variables introduced is $O(n_i h_j)$ for the sequential counters encoding, leading to large SAT representations. Hence, dense matrices lead to large encodings. Next, consider the logical structure of Equation 4. Note that we have an *equivalence relation* between the value of the neuron and the cardinality constraint. A key component of a SAT solver is the inference procedure, that at each point of the search infers values of unassigned variables. Suppose Equation 4 produces the following constraint $\mathbf{x}_1^1 \Leftrightarrow (\bar{\mathbf{x}}_0^0 + \bar{\mathbf{x}}_1^0 + \mathbf{x}_2^0 \geq 2)$. Moreover, suppose $\mathbf{x}_0^1$ and $\mathbf{x}_2^0$ have not been fixed yet and $\mathbf{x}_0^0 = 1$. Note that any setting of variables $\mathbf{x}_1^0, \mathbf{x}_2^0$ can be extended to a valid assignment in this constraint. Hence, no inference is possible. In contrast, assume we only had cardinality constraints, i.e. $\bar{\mathbf{x}}_0^0 + \bar{\mathbf{x}}_1^0 + \mathbf{x}_2^0 \geq 2$, then we can infer that $\mathbf{x}_1^0 = 0$ and $\mathbf{x}_2^0 = 1$ at this point. As the vast majority of constraints in the encoding contain the equivalence relation, this hinders the inference algorithm's ability to find implied variables, making the solver dive deep before a conflict occurs.

---

[2]The error bounds depend on the parameters of the model counting algorithm.

[3]For example, dealing with high-precision floating-point arithmetic requires specialized tools.

**Block level** Next we go up one level of granularity and focus on the encoding of BLOCK. We note that a block is encoded by a set of constraints over the same variables. We count over these variables therefore multiple times. This observation hints at an opportunity to identify shared computations and exploit them to get more succinct encodings. For example, if we have two constraints $(\mathbf{x}_0^0 - \mathbf{x}_1^0 + \mathbf{x}_2^0 \geq 2)$ and $(-\mathbf{x}_0^0 - \mathbf{x}_1^0 + \mathbf{x}_2^0 \geq -2)$ then we encode the partial sum $-\mathbf{x}_1^0 + \mathbf{x}_2^0$ for both constraints and this can be shared. Our ability to share computations depends on the patterns in which 1 and $-1$ appear in the matrix $A$ as these coefficients control whether variables appear positively or negatively in each constraint. As $A$ is a full matrix and *no patterns are enforced* on occurrences of 1 and $-1$ between rows the amount of sharing is rather small for the standard BNN architecture.

**Network level** Finally, we consider the entire BNN as a chain of blocks. As each block is encoded individually, BINBNN is a conjunction of BINBLOCK formulas that are loosely connected via variables $\mathbf{x}^i$, $i = 1, \ldots, d-1$. As discussed above, $\mathbf{x}^i$ variables are a small fraction of the total number of variables. Two factors affect solver performance. First, the block-wise structure of the BINBNN formula suggests that *formula decomposition* can be exploited by the solver. Second, the BINBNN formula effectively simulates the network function for all possible inputs. Additional constraints on inputs/outputs of the network, e.g., in the verification problem in Section 3.2, are the main source of inconsistencies. Therefore, conflict-driven learning, which is the core of all SAT solvers, should perform *directed search*, starting from either the last or the first blocks of the formula moving toward the first (last) blocks as search progresses rather than jumping unguided between layers. Such guidance might help to discover conflicts faster.

## 5 TOWARD SAT-FRIENDLY BNNS

We discuss how alternative architectural design choices for BNNs can be used to take advantage of the identified properties to reason faster. We observe that, assuming a fixed network architecture, our design options are limited. We therefore make an important assumption that we can change the architecture of the BNN and have access to the training procedure. While this is a strong assumption, we believe that BNN designers will be willing to incorporate changes into the training procedure as long as they do not cause accuracy loss. As before, we structure our solutions and observations using different levels of granularity.

**Neuron level** As discussed above, there are two properties of Equation 4 that need to be addressed. First issue is that $A$ is a dense matrix by design of BNNs. To deal with this issue, we propose using a ternary quantization procedure instead of a binary quantization to learn sparse matrices for linear transformations. We recall that BNN is trained using two matrices: a full-precision matrix $W^i$ that is updated during the gradient decent and a binarized matrix $A^i$, connected as follows $A^i = \text{sign}(W^i)$. $A^i$ is used on the forward path and during inference. The modification we make is a simple two-sided ternary quantization of $A^i$ during training:

$$a_{jt}^i = \begin{cases} sign(w_{jt}^i) & \text{if } \mid w_{jt}^i \mid \geq T, \\ 0 & \text{otherwise,} \end{cases} \tag{5}$$

where $T$ is a parameter. In principle, $T$ can be a learnable parameter (Zhu et al., 2017). However, in (Darabi et al., 2018), it was observed that weights of $W$ are naturally grouped around points $-1, 1$ and $0$ during training. Therefore, we use a fixed threshold $T$, obtained from the distribution of weights of a trained vanilla BNN. Allowing zero weights in $A$ requires some modifications in the original encoding of the last layer that we discuss in Appendix B.1. Consider how ternary quantization helps to reduce the encoding in our example.

**Example 5.1.** *Consider Example 3.1. We recall that the first row of $A$ is $[1, -1, 1]$ and the corresponding reified cardinality constraint is $\mathbf{x}_0^1 \Leftrightarrow (\mathbf{x}_0^0 - \mathbf{x}_1^0 + \mathbf{x}_2^0 \geq 1/2)$, (see Example 3.2). Now we assume that after using ternary quantization the first row of $A$ is $[1, 0, 0]$. Then the last reified cardinality constraint is simplified to $\mathbf{x}_0^1 \Leftrightarrow (\mathbf{x}_0^0 \geq 1)$, which is mush simpler constraint.* □

Interestingly, our approach resembles the lottery ticket hypothesis training framework introduced by Frankle & Carbin (2019). The main conceptual difference is that pruning and training are tightly coupled in our case as ternary quantization of weights is performed during every forward path. In Frankle & Carbin (2019), training and pruning are two separate phases[4].

---

[4]We thank an anonymous reviewer for pointing out the connection to (Frankle & Carbin, 2019)

The second issue at this level is that SAT solver inference is weak due to equivalence relations in Equation 4. Here we take the stabilization approach proposed in (Xiao et al., 2019) and adjust it to work for BNNs. The idea is to perform bounds propagation of the input bounds through the network to estimate lower and upper bounds of each neuron in the network. We denote $z_j$ an input of the sign function of a neuron $x_j^{i+1}$, $x_j^{i+1} = \mathrm{sign}(z_j)$ (Equation 1 defines $z_j$). If $\mathrm{sign}(ub(z_j)) = \mathrm{sign}(lb(z_j))$ then the sign operator is stable in the sense that we know the value of $x_j^{i+1}$, e.g. if both bounds are positive then $x_j^{i+1} = 1$, otherwise $-1$. For stable neurons, we can remove the equivalence relation and fix $x_j^{i+1}$ simplifying the encoding (see Appendix C for more details on bounds propagation and an example).

Estimates of lower and upper bounds must be binarized as the bounds computation goes across BLOCKs. To mimic the computation flow on the bounds computation path, we apply *hard* tanh before the sign operator: $lb(x_j^{i+1}) = \mathrm{sign}(\textsc{HardTanh}(lb(z_j)))$ and $ub(x_j^{i+1}) = \mathrm{sign}(\textsc{HardTanh}(ub(z_j)))$. We must also take into account that parameters of the BN layer are learned in a different way. Namely, $\mu_j^i$ and $\sigma_j^i$ are computed from the batch of samples, while other parameters are learned via gradient descent (Ioffe & Szegedy, 2015b). When we compute bounds, we pass them through the BN layer, but we should exclude these fake inputs from the computation of $\mu_j^i$ and $\sigma_j^i$. Finally, to achieve the effect of stabilization during training, for each neuron, we encourage $ub(z_j))$ and $lb(z_j)$ to take the same sign by adding an extra term to the loss function that approximates $\mathrm{sign}(ub(z_j))\,\mathrm{sign}(lb(z_j))$ as $-\tanh(1 + ub(z_j)lb(z_j))$ (Xiao et al., 2019).

**Block level**  We recall that the main issue we identified is that it is not obvious how to take advantage of shared partial computations in constraints of a block. One solution could be to enforce a pattern of zero and non-zero coefficients for each matrix $A$. A recent and successful example of this is a butterfly matrix (Dao et al., 2019b;a). This is a structured matrix with a predefined sparsity pattern. The original matrix is replaced with a composition of highly sparse matrices of a rigid structure that provides a very good approximation of the original matrix (Dao et al., 2019a). This new matrix can potentially be used to create a natural decomposition of our block constraints. However, it is a matter of future research if butterfly matrices can be used alongside quantization. Another solution that we can use is a different binary quantization, e.g. $-1/0$ or $0/1$, for each row of the matrix instead of ternary quantization. This will ensure that variables either occur in the negative/positive polarity or are absent in each constraint. Keeping the same polarity per constraint may foster shared computations. However, the distribution of zeroes is also important here to get a good level of sharing. Hence, we believe that using predefined patterns might be more promising.

**Network level**  Recall that we found two avenues for optimizing encoding at the network level: *formula decomposition* and *directed search*. In (Narodytska et al., 2018; Cheng et al., 2018), decomposability of BINBNN was exploited in limited forms there the authors independently proposed using CEGAR-like search to reason about property verification. (Khalil et al., 2019) took advantage of both decomposition and directed search by proceeding from the last to the first layer for a similar problem. However, while all these ideas appear to be helpful, there is no solver designed to incorporate such techniques. On the other hand, a number of verification frameworks exploit similar domain properties for bounded model checking (Bradley, 2011; Gurfinkel et al., 2015). So, it would be interesting to see if these solvers can be adapted for property verification of ML models.

# 6  EXPERIMENTS

We focus on the improvements we proposed for the neuron level of granularity. We use three datasets from (Narodytska et al., 2018; Baluta et al., 2019; Khalil et al., 2019). For each, we experiment with two tasks. First, we check if there is an untargeted adversarial attack (Goodfellow et al., 2015). Second, we compute an approximate number of adversarial examples for a given image. We used 4 hidden layers and one output layer. The input of the network is an image of size $28 \times 28$. The network has the following dimensions for all layers: $[784, 500, 300, 200, 100, 10]$. This gives $623K$ parameters, which is 3 to 10 times bigger than the networks in (Narodytska et al., 2018; Baluta et al., 2019). We used a full-precision trained network to seed weights before binarized training (Alizadeh et al., 2019). As the first layer inputs are reals, we used an additional BN + sign layer after the

| BNNs | MNIST | | FASHION | | MNISTBG | |
|---|---|---|---|---|---|---|
| | % | #prms | % | #prms | % | #prms |
| Vanilla | 96.5 | $623K$ | 82.1 | $623K$ | 74.3 | $623K$ |
| Sparse | 96.4 | $32K$ | 84.1 | $37K$ | 78.2 | $41K$ |
| Sparse+Stable | 95.9 | $32K$ | 83.2 | $37K$ | 78.3 | $38K$ |
| Sparse+L1 | 96.0 | $20K$ | 83.7 | $35K$ | 78.4 | $36K$ |
| Sparse+L1+Stable | 95.2 | $20K$ | 82.9 | $37K$ | 80.0 | $34K$ |

Table 2: The test accuracy (%) and the number of non-zero parameters (#prms) of the trained networks.

input layer to binarize inputs (see Appendix D.1). We used the PySAT tool (Ignatiev et al., 2018) to encode logical constraints to SAT and Glucose as a SAT solver (Audemard & Simon, 2018).

**Untargeted adversarial attacks.** In this task, we look for a perturbation $x'$ of the given input $x$ where $|x - x'| \leq \epsilon$. We used five values of epsilon, $\epsilon \in \{1, 3, 5, 10, 15, 20\}$[5]. We train three groups of networks. First, we trained vanilla BNNs. Then we trained two types of BNNs using our ideas from the neuron level. We trained BNNs with the ternary quantization method ('Sparse'). We used the value of $T = 0.03$ for MNISTBG and FASHION and $T = 0.04$ for MNIST using 60%-80% quantile depending on the dataset and the accuracy of the resulting network. Finally, we trained BNNs with stabilization of neurons ('Sparse+Stable'). We seeded initial weights of 'Sparse+Stable' network with weights of the trained 'Sparse' network as such a setup up gave better accuracy. In addition, we investigate the effect of a popular L1 regularizer on ternary quantization during the training ('Sparse+L1') and it stabilization version ('Sparse+L1+Stable').

Table 2 summarizes test accuracy results and the number of non-zero parameters per network for all trained networks and datasets. From Table 2, we can see that we can significantly reduce the number of parameters using ternary quantization during the training ('Sparse'). Interestingly, we do not lose test accuracy compared to the vanilla BNNs in all cases. If we consider the effect of L1 regulazation then we see that the number of parameters reduces even further but the accuracy drops a bit as well: the more we reduce the number of non-zero parameters the more we lose in terms of the accuracy. Using additional stabilization can also hurt the accuracy a bit for both 'Sparse' and 'Sparse+L1' networks. However, on average, we get about 40% of signs stabilized.

Figure 1 shows the performance of a SAT solver on three datasets. Results are averaged over 100 benchmarks. The average size of the encoding for all datasets and $\epsilon = 5$ are shown in Table 4. Note that the size of the encoding is 50x smaller compared to millions of variables and classes reported in Narodytska et al. (2018) for MNIST and MNISTBG for a much smaller network and the same $\epsilon$. Another interesting observation is that stabilization helps a lot to reduce the size of the encoding. If the PySAT tool takes more than 10G of memory to generate a SAT encoding then we terminate the generation process. If a solver time/memory outs on more than 70% of benchmarks we do not plot these results. Therefore, we do not have a plot for vanilla BNNs in these datasets as we either had a timeout or (mostly) memory out exceptions. Each plot shows the average time to solve the untargeted adversarial attack problem for each value of $\epsilon$. In addition, Figure 3 shows the number of solved problems for each $\epsilon$ value within the timeout (100 sec) by the SAT solver. As can be seen from these plots, in all except one case, at least 95% of benchmarks were solved within the 100 sec timeout

The plots show that ternary quantization greatly improves performance on three datasets as we were not even able to solve any benchmarks for the vanilla case. Using the 'L1' regularizer slightly improves performance in two datasets and helps a lot on MNIST. Finally, the stabilization of signs method consistently improves performance of a SAT solver across datasets for both 'Sparse' and 'Sparse+L1' networks, e.g. we get from 3x to 6x speedup due to stabilization.

Figure 2 shows the number of successful attacks as it is an interesting measure to understand properties of a network. Each plot shows the ratio of successful adversarial attacks for each value of $\epsilon$. Naturally, this ration plateaus at 1 as the value of perturbation $\epsilon$ grows. The results are mixed here. In some cases, using 'L1' does not significantly change the vulnerability of the network compare to 'Sparse' network, e.g. FASHION and MNISTBG, while it does help for MNIST. Interestingly, stabilization of signs can lead to increase in vulnarabilities if we compare a network with and with-

---

[5]We are able to use large values of $\epsilon$ compared to Narodytska et al. (2018), where $\epsilon \leq 5$.

out stabilization. Consider for example, MNIST with $\epsilon = 5$. Only 13 out of 100 benchmarks were successfully attacked for 'Sparse+L1' while 43 were attacked for 'Sparse+L1+Stable'.

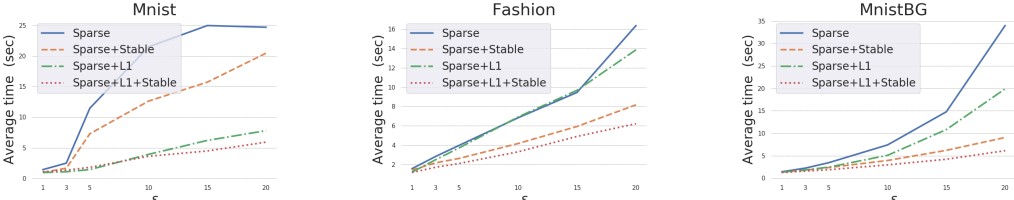

Figure 1: Performance of a SAT solver on the untargeted adversarial attack task.

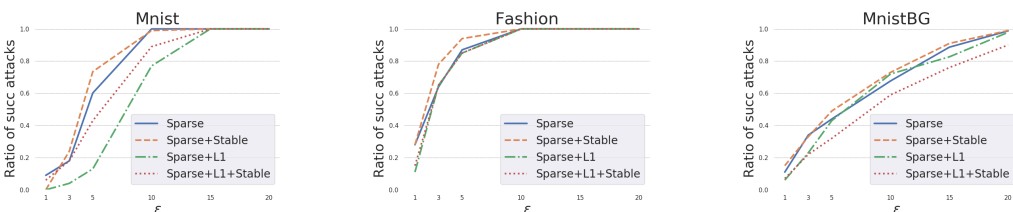

Figure 2: The percentage of successful adversarial attacks.

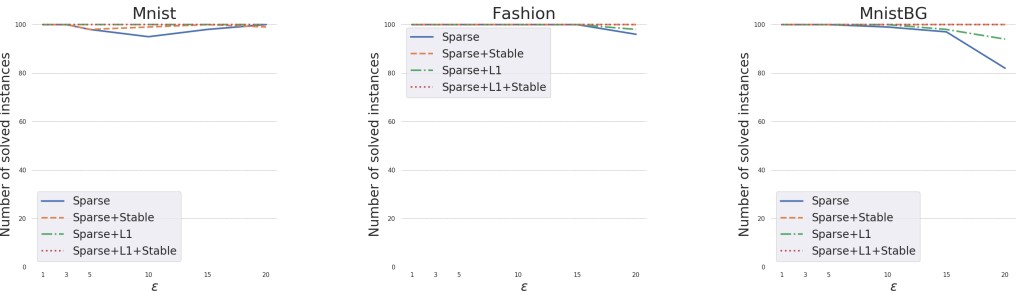

Figure 3: The number of solved benchmarks.

Finally, Table 3 shows original (the top row) and perturbed (the bottom row) samples of images from MNISTBG dataset for the following values of $\epsilon = 3, 5, 15, 20$ (from left to right). As can be seen from these sample, if $\epsilon = 3$ then original and perturbed images are identical. For $\epsilon = 20$ (last column), perturbations are visible.

**Likelihood of adversarial examples.** Next we estimate the probability of a perturbation to be an adversarial example. We reproduce a setup from (Baluta et al., 2019) where an approximate model counting, ApproxMC3, was used to perform such estimate. The main goal of this experiment is to demonstrate that our approach allows approximate model counting techniques to scale to larger networks. We focus on robustness as above, so SAT formulas check for existence of untargeted adversarial attacks. Hence, each solution corresponds to an adversarial attack. We perform model counting on binarized inputs (see Appendix D.1) with a timeout of 600 sec. We use default parameters of ApproxMC3. Table 4 summarizes results. We report the mean value of the solution count estimate divided by the total number of solutions (P(adv)) and the percentage of benchmarks solved by ApproxMC3 out of all benchmarks that we know are susceptible to adversarial perturbation (#solved). On average, we can solve 80% of benchmarks within the timeout. This contrasts with results reported in (Baluta et al., 2019), where it took 24 *hours* to perform approximate model counting for a BNN with about 50K parameters and a large fraction of benchmarks were not solved. Table 4 shows that the probability of adversarial attack is small for two out of three datasets. Another observation is sparsification and stabilization do not significantly change the probability of an input to be adversarial.

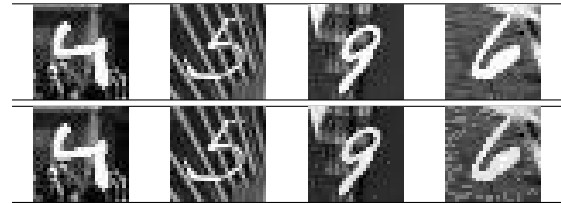

Table 3: The original (the top row) and successfully perturbed (the bottom row) images for MNISTBG for $\epsilon = 3, 5, 15, 20$.

| BNNs | MNIST | | | FASHION | | | MNISTBG | | |
|---|---|---|---|---|---|---|---|---|---|
| | #vars/#cls | P(adv) | #solved | #vars/#cls | P(adv) | #solved | #vars/#cls | P(adv) | #solved |
| Sparse | 63K/224K | 0.04 | 71% | 34K/116K | 0.04 | 87% | 24K/80K | 0.17 | 95% |
| Sparse+Stable | 42K/146K | 0.05 | 70% | 19K/58K | 0.06 | 89% | 12K/36K | 0.16 | 93% |
| Sparse+L1 | 8K/20K | 0.07 | 100% | 34K/115K | 0.07 | 87% | 17K/53K | 0.17 | 98% |
| Sparse+Stable+L1 | 11K/33K | 0.05 | 95% | 12K/33K | 0.06 | 98% | 10K/28K | 0.16 | 94% |

Table 4: The average size of the SAT encoding in terms of the number of variables (#vars) and clauses (#cls), the probability of perturbation to be an adversarial attack(P(adv)) and the number of solved benchmarks out all of benchmarks that can be attacked for $\epsilon = 5$.

**A ILP solver performance.** For completeness of the evaluation, we report results of an ILP solver on the same benchmarks for the untargeted adversarial attacks. The second experiment (to estimate the probability of a perturbation to be an adversarial example) is not currently possible for ILP formulations as we are not aware of an efficient approximate model counting solver for ILPs. We used the CPLEX solver with default configuration. Figure 4 shows results. As before Figure 4 shows the performance of a ILP solver on the untargeted adversarial example task for each value of $\epsilon$. Results are averaged over 100 benchmarks. We observe similar patterns as for ILP solvers: sparsification and stabilization are mostly helpful for all datasets as we cannot solve most of the benchmarks if we do not apply these techniques. As can be seen from the graph, the ILP solver exhibits a bell shape in two datasets, MNIST and FASHION: if $\epsilon$ is small or large then the instance is easier to solve. $\epsilon$ values in the middle are the hardest to solve for an ILP solver. We have not observed this bell shape for SAT solvers. Overall, if we compare performance of SAT and ILP solvers, the SAT solver is more efficient and solves more benchmarks (see Figure 5 for the number of solved benchmarks in Appendix D.2). In summary, our results show that a significant speed

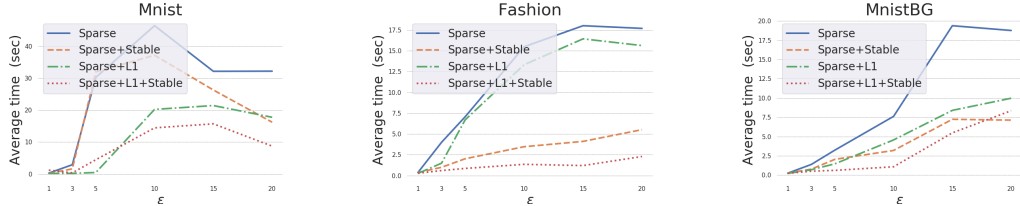

Figure 4: Performance of an ILP solver on the untargeted adversarial attack task.

up can be obtained if we design an architecture of BNNs and its training procedure taking into account potential bottlenecks of SAT solvers. We achieved 10-20x speed up in verification and in probabilistic queries compared to state of the art results in (Narodytska et al., 2018; Baluta et al., 2019).

# 7 CONCLUSIONS

In this work, we analyze architectural design choices of BNNs and discuss how they affect the performance of logic-based reasoners, focusing on the individual neuron and block levels. We demonstrated how to train BNNs so that the resulting network is easier to verify for logic-based engines, like SAT solver or approximate model counting solvers.

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

## A    BACKGROUND. ADDITIONAL DETAILS.

**Model counting.**    The problem of model counting is to calculate the number satisfying assignments or models of a given SAT formula. This problem is complete for the complexity class $\#\mathcal{P}$ (Valiant, 1979), which contains the entire polynomial hierarchy. Despite the high complexity, a number of tools for model counting have been developed (Sang et al., 2005; Thurley, 2006; Muise et al., 2012; Lagniez & Marquis, 2017), which were shown to work well on certain benchmarks arising in practice. However, for many applications, obtaining exact counts is not necessary and a good approximation often suffices, especially when it leads to better scalability. These requirements have led to the emergence of *approximate* counting approaches (Ermon et al., 2013; Chakraborty et al., 2013; Soos & Meel, 2019) which employ universal hash functions along with specialized SAT solvers (Soos et al., 2009) for balancing accuracy and scalability. The most successful approximate counters provide Probabilistically Approximately Correct (PAC)-style guarantees (Valiant, 1984) on accuracy, while scaling to formulas with thousands of variables in practice. In our experiments, we use a state-of-the-art tool called ApproxMC3 (Soos & Meel, 2019) which has been shown to scale much better than other exact and approximate counters. A key advantage of ApproxMC3 is that it supports *projected* model counting, i.e. counting over a subset of the variables in the given formula, which is essential for our purposes as we count over variables that represent inputs of the network.

**Sequential counters encoding.**    We recall the definition of *sequential counters* encoding from (Sinz, 2005a;b) that are used to encode cardinality constraints. Consider a cardinality constraint: $\sum_{i=1}^{m} l_i \geq k$, where $l_i \in \{0, 1\}$ is a Boolean variable, $\mathbf{x}_i$, or its negation, $\bar{\mathbf{x}}_i$, and $k$ is a constant. Then the sequential counter encoding, can be written as the following Boolean formula using auxiliary Boolean variables $r_{(j,k)}$:

$$(l_1 \Leftrightarrow r_{(1,1)}) \wedge (\bar{r}_{(1,j)}, \ \ j \in \{2, \ldots, k\}) \ \wedge$$
$$r_{(i,1)} \Leftrightarrow l_i \vee r_{(i-1,1)} \ \wedge$$
$$r_{(i,j)} \Leftrightarrow l_i \wedge r_{(i-1,j-1)} \vee r_{(i-1,j)}, j \in \{2, \ldots, k\}, \tag{6}$$

where $i \in \{2, \ldots, m\}$. Essentially, this encoding computes the cumulative sums $\sum_{i=1}^{k} l_i$ and the computation is performed in unary, i.e., the Boolean variable $r_{(j,k)}$ is true iff $\sum_{i=1}^{j} l_i \geq k$. In particular, $r_{(m,k)}$ encodes whether $\sum_{i=1}^{m} l_i$ is greater than or equal to $k$.

# B  PROPOSITION ENCODING OF BNNS.

## B.1  ENCODING OF THE OUTPUT LAYER AND ITS ADJUSTMENT FOR TERNARY QUANTIZATION

We recall that the OUTPUT block is just a linear transformation: $o = Cx^{d-1} + q$.

**Example B.1.** *We continue Example 3.1. Suppose our single block is followed by an output block with two inputs and two outputs. We define transformation parameters as follows.*

$$\text{LIN}: \ C = [1, -1; -1, -1], q = [0.5, -0.1] \tag{7}$$

*Hence, we can write the following system of linear constraints over $\{-1, 1\}$ variables.*

$$o_0 = x_0^1 - x_1^1 + 0.5,$$
$$o_1 = -x_0^1 - x_1^1 - 0.1.$$

*As before, we perform variable replacement to obtain constraints over Boolean variables only.*

$$o_0 = 2\mathbf{x}_0^1 - 1 - (2\mathbf{x}_1^1 - 1) + 0.5,$$
$$o_1 = -(2\mathbf{x}_0^1 - 1) - (2\mathbf{x}_1^1 - 1) - 0.1.$$

*After rearrangements, we get*

$$o_0 = 2\mathbf{x}_0^1 - 2\mathbf{x}_1^1 + 0.5,$$
$$o_1 = -2\mathbf{x}_0^1 - 2\mathbf{x}_1^1 + 1.9. \quad \square$$

Usually, we are not interested in absolute values of $o$. Rather, we need to know the index of the maximum element of $o$, which corresponds to the top prediction in the classification task. Hence, we need to encode the ordering relation over variables $o$. To do so, we use an additional set of Boolean variables $d_{ij}, i = 1, \ldots, p, j = i+1, \ldots, p$. Applying variable replacements and rearranging terms, we get:

$$d_{ij} \Leftrightarrow \left( \sum_{t=1}^{p} c_{it} 2\mathbf{x}_t^d - \sum_{t=1}^{p} c_{it} + q_i \geq \sum_{t=1}^{p} c_{jt} 2\mathbf{x}_t^d - \sum_{t=1}^{p} c_{jt} + q_j \right) \tag{8}$$

we denote $R_{ij} = (\sum_{t=1}^{p} c_{it} - q_i - \sum_{t=1}^{p} c_{jt} + q_j)$, so we get $\tag{9}$

$$\Leftrightarrow \left( \sum_{t=1}^{p} c_{it} \mathbf{x}_t^d - \sum_{t=1}^{p} c_{jt} \mathbf{x}_t^d \geq \frac{R_{ij}}{2} \right) \tag{10}$$

$$\Leftrightarrow \left( \sum_{t=1}^{p} (c_{it} - c_{jt}) \mathbf{x}_t^d \geq \frac{R_{ij}}{2} \right) \tag{11}$$

Finally, we note that $c_{it}, c_{jt} \in \{-1, 1\}$ are applied to the same variable $x_t^d$. Hence, depending on the sign of these coefficients, the difference $|c_{it} - c_{jt}|$ is either 0 or 2. So we can rewrite:

$$d_{ij} \Leftrightarrow \left( 2 \times \sum_{t=1, c_{it}-c_{jt}=2}^{p} \mathbf{x}_t^d - 2 \times \sum_{t=1, c_{it}-c_{jt}=-2}^{p} \mathbf{x}_t^d \geq \frac{R_{ij}}{2} \right) \tag{12}$$

$$\Leftrightarrow \left( \sum_{t=1, c_{it}-c_{jt}=2}^{p} \mathbf{x}_t^d - \sum_{t=1, c_{it}-c_{jt}=-2}^{p} \mathbf{x}_t^d \geq \left\lceil \frac{R_{ij}}{4} \right\rceil \right) \tag{13}$$

**Example B.2.** *Coming back to our Example B.1, as we have two variables, we introduce one Boolean variable $d_{01}$ to signal the top prediction:*

$$d_{0,1} \Leftrightarrow o_0 \geq o_1$$
$$\Leftrightarrow (2\mathbf{x}_0^1 - 2\mathbf{x}_1^1 + 2\mathbf{x}_0^1 + 2\mathbf{x}_1^1 \geq 1.9 - 0.5)$$
$$\Leftrightarrow (\mathbf{x}_0^1 - \mathbf{x}_1^1 + \mathbf{x}_0^1 + \mathbf{x}_1^1 \geq 1.4/2)$$
$$\Leftrightarrow \mathbf{x}_0^1 \geq \lceil 1.4/4 \rceil) \quad \square$$

The final observation is that if $C$ is a sparse matrix then we lose the property that the difference $c|_{it} - c_{jt}|$ is either 0 or 2 as $c_{it}, c_{jt} \in \{-1, 0, 1\}$. Hence, the last division by two (Equation 13) is not possible. However, we take advantage of the fact that coefficients are small, $|c_{it} - c_{jt}| \leq 2$. So, we can duplicate variables to model non-unary coefficients, for example.

**Example B.3.** *Suppose, we get the following matrix $C$ due to ternary quantization:*

$$\text{LIN}: \; C = [1, 0; -1, -1], q = [0.5, -0.1] \tag{14}$$

*Then we have the following ordering constraint:*

$$
\begin{aligned}
d_{0,1} &\Leftrightarrow o_0 \geq o_1 \\
&\Leftrightarrow (2\mathbf{x}_0^1 - 1 + 0.5 \geq -2\mathbf{x}_0^1 + 1 - 2\mathbf{x}_1^1 + 1 - 0.1) \\
&\Leftrightarrow (\mathbf{x}_0^1 + \mathbf{x}_0^1 + \mathbf{x}_1^1 \geq 2.4/2) \\
&\Leftrightarrow 2\mathbf{x}_0^1 + \mathbf{x}_1^1 \geq \lceil 2.4/2 \rceil)
\end{aligned}
$$

*We can introduce an extra variable $x_0^{1'}$ and replace $2\mathbf{x}_0^1 + \mathbf{x}_1^1$ with $\mathbf{x}_0^1 + \mathbf{x}_0^{1'} + \mathbf{x}_1^1$ and an equality constraint $\mathbf{x}_0^1 = \mathbf{x}_0^{1'}$.* □

## C  BOUNDS PROPAGATION

Let us split BLOCK computation into two parts that correspond to LIN and BN layers: $y_j = \sum_{t=1}^{n_i} a_{j,t}^i x_t^i + b_j^i$ and $z_j = \frac{\alpha_j^i}{\sigma_j^i}(y_j - \mu_j^i) + \gamma_j^i$ to demonstrate computations of lower and upper bounds on $y_j$ ($z_j$ are computed similarly). We get that

$$lb(y_j) = \sum_{t=1, a_{j,t}^i > 0}^{n_i} a_{j,t}^i lb(x_t^i) + \sum_{t=1, a_{j,t}^i < 0}^{n_i} a_{j,t}^i ub(x_t^i) + b_j^i \tag{15}$$

$$ub(y_j) = \sum_{t=1, a_{j,t}^i > 0}^{n_i} a_{j,t}^i ub(x_t^i) + \sum_{t=1, a_{j,t}^i < 0}^{n_i} a_{j,t}^i lb(x_t^i) + b_j^i \tag{16}$$

**Example C.1.** *Consider our example for BLOCK and the second neuron computation:*

$$x_1^1 = \text{sign}(0.1(-x_0^0 - x_1^0 + x_2^0 + 1) + 0.2 \geq 0).$$

*Suppose we derive that for all possible samples, $ub(x_0^0) = ub(x_1^0) = -1$. In this case, we have $lb(x_0^0) = lb(x_1^0) = -1$ as all activations are in $\{-1, 1\}$. We can compute bounds on the all variables $y$ and $z$.*

$$
\begin{aligned}
lb(y_1) &= lb(-x_0^0 - x_1^0 + x_2^0) = 1 + 1 - 1 = 1, \\
ub(y_1) &= ub(-x_0^0 - x_1^0 + x_2^0) = 1 + 1 + 1 = 3, \\
lb(z_1) &= 0.1 \times lb(y_1 + 1) + 0.2 = 0.4, \\
ub(z_1) &= 0.1 \times ub(y_1 + 1) + 0.2 = 0.6.
\end{aligned}
$$

*So, we get that $\text{sign}(lb(z_1)) = \text{sign}(ub(z_1))$, so we know that $x_1^1 = 1$ and we can use this information to reduce the encodings.* □

## D  EXPERIMENTAL EVALUATION.

### D.1  BINARIZATION OF INPUTS.

As we discussed in Section 6, we introduce an additional block before the first internal layer that mimics binarization of inputs. This layer constraints BN and sign operators only (hard $\tanh$ is used during the training). Consider how this layer works.

$$\mathbf{x}_t^0 \Leftrightarrow \text{sign}\left(\frac{\alpha_t}{\sigma_t}(x_t' + b_t - \mu_t) + \gamma_t\right),$$

$$x_t' \in [x_t - \epsilon, x_t + \epsilon].$$

First, we consider two extreme values of the expression that is an input of sign w.r.t. $\epsilon$. Suppose $\alpha_t \geq 0$ ($\alpha_t < 0$ is analogous).

$$max_t = \frac{\alpha_t}{\sigma_t}(x_t + \epsilon + b_t - \mu_t) + \gamma_t, \tag{17}$$

$$min_t = \frac{\alpha_t}{\sigma_t}(x_t - \epsilon + b_t - \mu_t) + \gamma_t \tag{18}$$

If $min_t \geq 0$ then we know that $\mathbf{x}_t^0 = 1$. If $max_j < 0$ then we know that $\mathbf{x}_t^0 = 0$. In other cases, $\mathbf{x}_t^0 \in \{0, 1\}$. Consider the reverse transformation. If $\mathbf{x}_j^0 = 0$ is a solution of a problem produced by the SAT solver then we can map back to $x_t = min_t$. Similarly, if $\mathbf{x}_t^0 = 1$ then $x_t = max_t$. As each transformation in Equation 17–Equation 18 is performed on a single input, we can build a valid input from a solution over Boolean variables.

### D.2 EXPERIMENTAL EVALUATION USING ILP SOLVERS. MISSING FIGURE.

Figure 5 shows the number of solved problems for each $\epsilon$ value within the timeout.

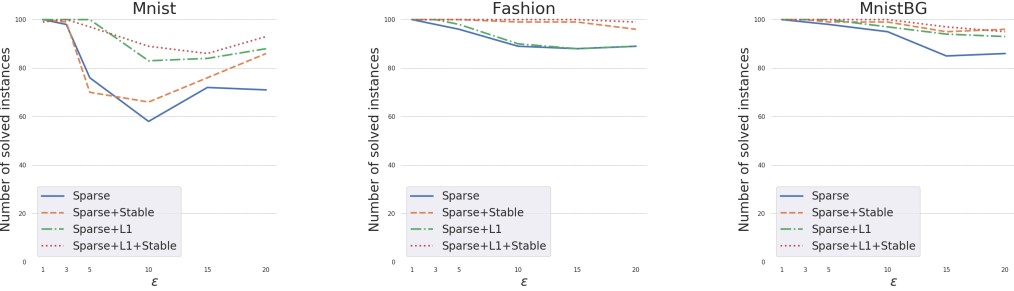

Figure 5: The number of solved benchmarks.

