# OpenReview forum: "In Search for a SAT-friendly Binarized Neural Network Architecture"
_ICLR.cc/2020/Conference — Accept (Poster)_

### Official Review · AnonReviewer2 · 2019-10-08
**Official Blind Review #2**

**Rating:** 6

**Review:**

In this paper, a new design of Binary Neural Network (BNN) is proposed. The purpose of the new design is to make the network easier to analyze using the existing Boolean satisfiability (SAT) solvers. An easier analysis is preferable since the SAT solver can be used to infer existing fragility in the network, e.g., adversarial attacks. Empirically, the proposed design of BNN is shown to outperform traditional BNNs by a large margin in terms of detecting adversarial attacks.

Overall, this paper seems to provide a solid contribution to the design of robust BNNs. I like how the authors tried to address various aspects of the networks for easy analysis of SAT solver, i.e., neuron, block, and network-levels even though they only focused on changing the architecture in neuron-level perspective. The experimental results seem to show benefits without any compromise in performance.



**Experience Assessment:**

I do not know much about this area.

**Review Assessment: Checking Correctness Of Derivations And Theory:**

I did not assess the derivations or theory.

**Review Assessment: Checking Correctness Of Experiments:**

I carefully checked the experiments.

**Review Assessment: Thoroughness In Paper Reading:**

I made a quick assessment of this paper.

---

> ### Author Response · Authors · 2019-11-08
> **Response**
>
> Thank you for your positive feedback!  We will upload an updated paper taking reviewers' suggestions into account in few days.

---

### Official Review · AnonReviewer3 · 2019-10-22
**Official Blind Review #3**

**Rating:** 6

**Review:**

This paper approaches the problem of using logic-based reasoning tools to analyze binary neural networks. They propose general changes to BNN architectures at the neuron level so as to adapt them for more efficient usage by such tools. Their main contributions are a new design choice utilizing ternary, rather than binary, quantization, and additional variables to propagate through the network containing upper and lower bounds for each layer output. They also suggest further promising directions to adapt BNNs at the block and network level. Their experiments show that the proposed changes do not significantly alter test accuracy from a standard trained baseline model while dramatically reducing the number of model parameters. Consequently, they are able to demonstrate that SAT solvers can find adversarial examples much more quickly with their new architectural alterations.
Overall, this paper presents a novel contribution with an idea for more efficient BNNs and experimentally verifies the success of their proposed changes. However, there is a concern regarding the experimental setup.
A large part of the motivation for this task arises from the field of verification regarding adversarial examples. While binarized neural networks are not my area of expertise, I am familiar with a number of verification papers on full neural networks. As such, I find the datasets used, accuracies reported, and model size a bit peculiar. In ‘Scaling provable adversarial defenses’ (Wong ’18), the authors were able to verify models with a couple million parameters on both MNIST and CIFAR. In my understanding, the primary advantage of BNNs is their efficiency in having only binary weights. If this is the case, I see no reason why the experimental setup would have such a small model. Additionally, it is generally well-known that MNIST and its variants are not a particularly hard datasets to classify. Many different architectures can easily achieve 98 or 99 percent test accuracy. A quick literature search found ‘A review of Binarized Neural Networks’ (Simons ’19), which reports better numbers on MNIST and decent results on CIFAR-10 (and in fact, even ImageNet). Their experimental evidence crucially relies on the conclusion that their techniques allow for faster SAT solver computation *while* maintaining comparable test accuracy. As such, it is my opinion that to be a strong paper, they must show this property holds for larger networks and more significant datasets.
Additionally, a comment for the authors to consider: I would suggest switching some of the supplementary materials with the main content. The exposition on BNNs is quite dense and detailed. Some of these details could be moved to the appendix and replaced with more experimental graphs and images. This would provide some breaks in the text so as to provide ease of reading.


**Experience Assessment:**

I do not know much about this area.

**Review Assessment: Checking Correctness Of Derivations And Theory:**

I carefully checked the derivations and theory.

**Review Assessment: Checking Correctness Of Experiments:**

I carefully checked the experiments.

**Review Assessment: Thoroughness In Paper Reading:**

I read the paper thoroughly.

---

> ### Author Response · Authors · 2019-11-08
> **Response**
>
> Thank you for your feedback and  detailed comments! Please find our answers below. We will upload an updated paper taking reviewers' suggestions into account in a few days.
>
> >> On comparison exact verification and certification approaches to verification, scalability and accuracy concerns.
>
> [Exact verification and certification methods].
> The work of 'Scaling provable adversarial defenses’[1] falls into the category of certification methods[1,6,7] that solve a relaxation of the verification problem.  Our work falls into the category of exact verification methods, similar to [2-4].  We argue that these two approaches are complementary and can be useful in different settings. We highlight few main properties that exact verification methods offer that certification methods cannot provide at the moment:
>
> a) accommodate a wide range of verification conditions while certification methods mostly focus on robustness.
>
> b) do not require knowledge of all properties during training, so exact verification methods decouples training and verification procedures (verification is performed after training is completed).
>
> c) can provide an exact answer to the verification question while certification methods cannot guarantee that they certify all robust inputs (as they are based on an over-approximation of adversarial polytope).
>
> d) [specific to BNNs]  BNNs admit a propositional logic representation, hence we can use techniques that go beyond existential questions that are mostly considered in the work on verification/robustness of NNs, e.g. probabilistic queries [5].
>
>
> [Scalability] As the reviewer pointed out, scalability is a weak point of all exact methods developed for full-precision or binarized networks. However, we believe that we make a significant step forward in the space of BNN networks compared to the existing work. In fact, we would like to point out that the certification methods line of work had to make few leaps to improve scalability to progress from small, two-layers networks in [6] to large networks in [1].  Making such leaps is even harder for exact methods as they have to provide exact guarantees. Therefore, we believe that our contribution is important as the network we verified is 3 times larger than any BNN-verified network reported in the literature (and we experimented with larger values of epsilons). Moreover, we achieved 10x speed up in verification and  a significant speed up for performing probabilistic queries related to security or fairness on BNN-based models [5], e.g. 12x compared to [5].
>
> [Why experimental setup has such a small model].
> We used the original implementation of BNNs from [8] (in PyTorch). Our accuracy is at least as high as that reported in [3-5] for network sizes of the same order of magnitude. As we mentioned above, our network is 3 times larger than any BNN-verified network in the literature.However, we agree that more work is needed to verify medium (for compute vision tasks) networks.
>
> We agree that it is possible to achieve near state-of-the-art accuracy with BNNs. For example, we would need 3 hidden layers of 4096 neurons in the BNN training framework[8]. This architecture gives a network of 38 million parameters. Such networks are still challenging for SAT/ILP-based technologies. To the best of our knowledge, none of exact verification tools can scale to networks with 38M parameters and 784 inputs for a reasonable epsilon value, e.g. we scaled for epsilon 20/256 in our experiments.
>
> >> Presentation suggestions.
>
> We will move the SAT encoding mostly to appendix to make space for more experiments in the main part in the updated version.
>
> [1] Eric Wong, Frank R. Schmidt, Jan Hendrik Metzen,  Zico Kolter:
> Scaling provable adversarial defenses. NeurIPS 2018
>
> [2] Guy Katz, Derek Huang, Duligur Ibeling, Kyle Julian, Christopher Lazarus, Rachel Lim, Parth
> Shah, Shantanu Thakoor, Haoze Wu, Aleksandar Zeljic, David L. Dill, Mykel Kochenderfer,
> and Clark Barrett.  The Marabou framework for verification and analysis of deep neural net-
> works,CAV'`19
>
> [3] Elias   Khalil,  Amrita  Gupta,  and  Bistra  Dilkina.  Combinatorial  attacks  on  binarized  neural networks, ICLR'19
>
> [4] Nina Narodytska, Shiva Prasad Kasiviswanathan, Leonid Ryzhyk, Mooly Sagiv, and Toby Walsh.
> Verifying properties of binarized deep neural networks, AAAI'18
>
> [5] Teodora Baluta, Shiqi Shen, Shweta Shinde, Kuldeep S. Meel, and Prateek Saxena.  Quantitative
> verification of neural networks and its security applications, CCS 2019.
>
> [6]Aditi Raghunathan, Jacob Steinhardt, and Percy Liang. Certified defenses against adversarial
> examples. ICLR'18
>
> [7] Zico Kolter, Eric Wong:
> Provable defenses against adversarial examples via the convex outer adversarial polytope. ICML'18
>
> [8] Binarized Neural Networks: Training Deep Neural Networks with Weights and Activations Constrained to +1 or -1 Matthieu Courbariaux, Itay Hubara, Daniel Soudry, Ran El-Yaniv, Yoshua Bengio, https://github.com/itayhubara/BinaryNet.pytorch

---

### Official Review · AnonReviewer4 · 2019-11-05
**Official Blind Review #4**

**Rating:** 8

**Review:**

This paper deals with the scalability of Binarized Neural Networks (BNNs) and their representation in Boolean logic. This encoding enables SAT solvers to reason about the underlying variables and query existential or counting clauses. The main contribution of this paper is the analysis of the architectural design choices of the BNN and modifications of the training procedure in order to improve the efficiency of SAT solvers. The modifications are simple but effective and the authors show consistent improvement on different benchmarks without the SAT solver timing out.

Overall, I am leaning towards accepting this paper due to the improved empirical results compared to the baselines the authors build on. However, it does not seem that there is much novelty in the BNN architecture per se, but rather in the training procedure. The modifications are simple but effective. The paper seems to be well-written and easy to follow also from a reader not familiar with the literature in this area.

I would be interested to have an answer to the following concerns:

- It was not totally clear for me what is are the changes that the paper proposes in the block level and network level and if these modifications are present in their experiments.
- Did you evaluate your method on other image classification datasets, such as CIFAR-10/-100?
- The thresholding used to sparsify the matrix A^i and seeding the initialization (correct me if I am wrong) when retraining the BNN seems to be really similar to the Lottery Ticket Hypothesis [1]. It would be interesting to evaluate the sensitivity of your proposed method towards the initialization.

[1] Jonathan Frankle, Michael Carbin. The Lottery Ticket Hypothesis: Finding Sparse, Trainable Neural Networks. In ICLR, 2019

**Experience Assessment:**

I do not know much about this area.

**Review Assessment: Checking Correctness Of Derivations And Theory:**

I assessed the sensibility of the derivations and theory.

**Review Assessment: Checking Correctness Of Experiments:**

I assessed the sensibility of the experiments.

**Review Assessment: Thoroughness In Paper Reading:**

I read the paper at least twice and used my best judgement in assessing the paper.

---

> ### Author Response · Authors · 2019-11-08
> **Response**
>
> Thank you for your positive feedback and detailed comments! Please find our answers below.  We will upload an updated paper taking reviewers' suggestions into account in a few days.
>
> >> It was not totally clear for me what is are the changes that the paper proposes in the block level and network level and if these modifications are present in their experiments.
>
> For the block and network levels, we only analyse bottlenecks, propose possible research directions, and position existing work on the network level. Our main contribution is within the network level.
>
> We would like to highlight that systematic level-based analysis of the network structure from the decision procedures standpoint is also a contribution of our work. It helps solver developers to understand where potential bottlenecks might come from.
>
> We stated these points in our contributions, but we will make it more clear in the revised version.
>
> >>  Did you evaluate your method on other image classification datasets, such as CIFAR-10/-100?
>
> We have not tried these networks. We would need much larger convolutional networks to get a reasonable accuracy BNN for CIFAR-10. We point out that these networks and datasets also were not considered in previous work on complete verification methods that used MNIST and FashionMNIST[1,2].
>
> [1] Elias  B.  Khalil,  Amrita  Gupta,  and  Bistra  Dilkina.  Combinatorial  attacks  on  binarized  neural networks, ICLR'19
>
> [2] Nina Narodytska, Shiva Prasad Kasiviswanathan, Leonid Ryzhyk, Mooly Sagiv, and Toby Walsh.
> Verifying properties of binarized deep neural networks, AAAI'18
>
> >>> The thresholding used to sparsify the matrix A^i and seeding the initialization (correct me if I am wrong) when retraining the BNN seems to be really similar to the Lottery Ticket Hypothesis [3]. It would be interesting to evaluate the sensitivity of your proposed method towards the initialization.
>
> Thank you for pointing out the similarity with [3]!  We agree that at a high level, we take the same approach that alternates training  and pruning of weights phases. The main conceptual difference is that pruning and training are tightly coupled in our case. Namely, on each forward path we perform -1/0/1 quantization of weights. In [1], the authors train a network until convergence, then prune weights and repeat. So, training and pruning are two separate phases.  We will discuss this connection in the paper.
>
> We have not investigated sensitivity to initialization so far. We start with training a full precision network to seed weights before binarized training [4]. We agree that it is interesting to look into sensitivity to the initial trained  full precision network as future work.
>
> [3] The Lottery Ticket Hypothesis: Finding Sparse, Trainable Neural Networks Jonathan Frankle, Michael Carbin
> [4] Milad Alizadeh, Javier Fernandez-Marques, Nicholas D. Lane, and Yarin Gal.  An empirical study of binary neural networks optimisation,   ICLR 2019.

---

> > ### Author Response · Authors · 2019-11-13
> > **Regarding sensitivity to initialization**
> >
> > We ran few experiments with MNIST with background to investigate sensitivity  to different initializations.
> > 1. We trained 5 full-precision networks.
> > 2. We trained 5 quantized (-1/0/1) networks seeding from these trained full-precision networks.
> > We observe a small variation (about 10%) in the number of non-zero parameters after quantized training.

---

> > > ### Comment · AnonReviewer4 · 2019-11-14
> > > **Response to Authors rebuttal**
> > >
> > > Thank you very much for taking my listed points into account.
> > >
> > > I think the paper is very well-written and the empirical results show clearly improved results towards baselines. In general, I think these type of papers that go towards the direction of having more interpretable NN and understanding the working mechanisms in detail are useful for the research community. Therefore, I will increase my score to 8 and vote for acceptance.

---

### Author Response · Authors · 2019-11-13
**Updated version**

Thank you again all reviewers for your comments!

We  uploaded an updated version of the paper:
*Made it clear that our main technical contribution is in the neuron layer (Page 2).
*Highlighted  a connection with the lottery ticket hypothesis paper suggested by reviewer 4 (Page 5).
*Revised the main part of the paper and moved BNN to SAT conversion to Appendix. We moved experimental results with ILP solvers back to the main part as reviewer 3 suggested.

---

### Decision · Program_Chairs · 2019-12-19

**Decision:**

Accept (Poster)

**Comment:**

This paper studies how the architecture and training procedure of binarized neural networks can be changed in order to make it easier for SAT solvers to verify certain properties of them.

All of the reviewers were positive about the paper, and their questions were addressed to their satisfaction, so all reviewers are in favor of accepting the paper. I therefore recommend acceptance.